# SQUEEZE TRAINING FOR ADVERSARIAL ROBUSTNESS

**Qizhang Li**[1,2], **Yiwen Guo**[3*], **Wangmeng Zuo**[1*], **Hao Chen**[4]
[1]Harbin Institute of Technology, [2]Tencent Security Big Data Lab, [3]Independent Researcher, [4]UC Davis
{liqizhang95,guoyiwen89}@gmail.com  wmzuo@hit.edu.cn  chen@ucdavis.edu

## ABSTRACT

The vulnerability of deep neural networks (DNNs) to adversarial examples has attracted great attention in the machine learning community. The problem is related to non-flatness and non-smoothness of normally obtained loss landscapes. Training augmented with adversarial examples (*a.k.a.*, adversarial training) is considered as an effective remedy. In this paper, we highlight that some collaborative examples, nearly perceptually indistinguishable from both adversarial and benign examples yet show extremely lower prediction loss, can be utilized to enhance adversarial training. A novel method is therefore proposed to achieve new state-of-the-arts in adversarial robustness. Code: https://github.com/qizhangli/ST-AT.

## 1  INTRODUCTION

Adversarial examples (Szegedy et al., 2013; Biggio et al., 2013) crafted by adding imperceptible perturbations to benign examples are capable of fooling DNNs to make incorrect predictions. The existence of such adversarial examples has raised security concerns and attracted great attention.

Much endeavour has been devoted to improve the adversarial robustness of DNNs. As one of the most effective methods, adversarial training (Madry et al., 2018) introduces powerful and adaptive adversarial examples during model training and encourages the model to classify them correctly.

In this paper, to gain a deeper understanding of DNNs, robust or not, we examine the valley of their loss landscapes and explore the existence of collaborative examples in the $\epsilon$-bounded neighborhood of benign examples, which demonstrate extremely lower prediction loss in comparison to that of their neighbors. Somewhat unsurprisingly, the existence of such examples can be related to the adversarial robustness of DNNs. In particular, if given a model that was trained to be adversarially more robust, then it is less likely to discover a collaborative example. Moreover, incorporating such collaborative examples into model training seemingly also improves the obtained adversarial robustness. On this point, we advocate squeeze training (ST), in which adversarial examples and collaborative examples of each benign example are jointly and equally optimized in a novel procedure, such that their maximum possible prediction discrepancy is constrained. Extensive experimental results verify the effectiveness of our method. We demonstrate that ST outperforms state-of-the-arts remarkably on several benchmark datasets, achieving an absolute robust accuracy gain of $>$**+1.00%** without utilizing additional data on CIFAR-10. It can also be readily combined with a variety of recent efforts, *e.g.*, RST (Carmon et al., 2019) and RWP (Wu et al., 2020b), to further improve the performance.

## 2  BACKGROUND AND RELATED WORK

### 2.1  ADVERSARIAL EXAMPLES

Let $\mathbf{x}_i$ and $y_i$ denote a benign example (*e.g.*, a natural image) and its label from $S = \{(\mathbf{x}_i, y_i)\}_{i=1}^n$, where $\mathbf{x}_i \in \mathcal{X}$ and $y_i \in \mathcal{Y} = \{0, \dots, C - 1\}$. We use $\mathcal{B}_\epsilon[\mathbf{x}_i] = \{\mathbf{x}' \mid \|\mathbf{x}' - \mathbf{x}_i\|_\infty \le \epsilon\}$ to represent the $\epsilon$-bounded $l_\infty$ neighborhood of $\mathbf{x}_i$. A DNN parameterized by $\Theta$ can be defined as a function $f_\Theta(\cdot) : \mathcal{X} \to \mathbb{R}^C$. Without ambiguity, we will drop the subscript $\Theta$ in $f_\Theta(\cdot)$ and write it as $f(\cdot)$.

In general, adversarial examples are almost perceptually indistinguishable to benign examples, yet they lead to arbitrary predictions on the victim models. One typical formulation of generating an

---

[*]Work was done under co-supervision of Yiwen Guo and Wangmeng Zuo who are in correspondence.

adversarial example is to maximize the prediction loss in a constrained neighborhood of a benign example.

Projected gradient descent (PGD) (Madry et al., 2018) (or the iterative fast gradient sign method, *i.e.*, I-FGSM (Kurakin et al., 2017)) is commonly chosen for achieving the aim. It seeks possible adversarial examples by leveraging the gradient of $g = \ell \circ f$ w.r.t. its inputs, where $\ell$ is a loss function (*e.g.*, the cross-entropy loss $\mathrm{CE}(\cdot, y)$). Given a starting point $\mathbf{x}^0$, an iterative update is performed with:

$$\mathbf{x}^{t+1} = \mathbf{\Pi}_{\mathcal{B}_\epsilon[\mathbf{x}]}(\mathbf{x}^t + \alpha \cdot \mathrm{sign}(\nabla_{\mathbf{x}^t}\mathrm{CE}(f(\mathbf{x}^t), y))), \tag{1}$$

where $\mathbf{x}^t$ is a temporary result obtained at the $t$-th step and function $\mathbf{\Pi}_{\mathcal{B}_\epsilon[\mathbf{x}]}(\cdot)$ projects its input onto the $\epsilon$-bounded neighborhood of the benign example. The starting point can be the benign example (for I-FGSM) or its randomly neighbor (for PGD).

Besides I-FGSM and PGD, the single-step FGSM (Goodfellow et al., 2015), C&W's attack (Carlini & Wagner, 2017), DeepFool (Moosavi-Dezfooli et al., 2019), and the momentum iterative FGSM (Dong et al., 2018) are also popular and effective for generating adversarial examples. Some work also investigates the way of generating adversarial examples without any knowledge of the victim model, which are known as black-box attacks (Papernot et al., 2017; Chen et al., 2017; Ilyas et al., 2018; Cheng et al., 2019; Xie et al., 2019; Guo et al., 2020) and no-box attacks (Papernot et al., 2017; Li et al., 2020). Recently, the ensemble of a variety of attacks becomes popular for performing adversarial attack and evaluating adversarial robustness. Such a strong adversarial benchmark, called AutoAttack (AA) (Croce & Hein, 2020), consists of three white-box attacks, *i.e.*, APGD-CE, APGD-DLR, and FAB (Croce & Hein, 2019), and one black-box attack, *i.e.*, the Square Attack (Andriushchenko et al., 2019). We adopt it in experimental evaluations.

In this paper, we explore valley of the loss landscape of DNNs and study the benefit of incorporating collaborative examples into adversarial training. In an independent paper (Tao et al., 2022), hypocritical examples were explored for concealing mistakes of a model, as an attack. These examples also lied in the valley. Yet, due to the difference in aim, studies of hypocritical examples in (Tao et al., 2022) were mainly performed based on mis-classified benign examples according to their formal definition, while our work concerns local landscapes around all benign examples. Other related work include unadversarial examples (Salman et al., 2021) and assistive signals (Pestana et al., 2021) that designed 3D textures to customize objects for better classifying them.

## 2.2 ADVERSARIAL TRAINING (AT)

Among the numerous methods for defending against adversarial examples, adversarial training that incorporates such examples into model training is probably one of the most effective ones. We will revisit some representative adversarial training methods in this subsection.

**Vanilla AT** (Madry et al., 2018) formulates the training objective as a simple min-max game. Adversarial examples are first generated using for instance PGD to maximize some loss (*e.g.*, the cross-entropy loss) in the objective, and then the model parameters are optimized to minimize the same loss with the obtained adversarial examples:

$$\min_\Theta \max_{\mathbf{x}'_i \in \mathcal{B}_\epsilon[\mathbf{x}_i]} \mathrm{CE}(f(\mathbf{x}'_i), y_i). \tag{2}$$

Although effective in improving adversarial robustness, the vanilla AT method inevitably leads to decrease in the prediction accuracy of benign examples, therefore several follow-up methods discuss improved and more principled ways to better trade off clean and robust accuracy (Zhang et al., 2019; Kannan et al., 2018; Wang et al., 2020; Wu et al., 2020b). Such methods advocate regularizing the output of benign example and its adversarial neighbors. With remarkable empirical performance, they are regarded as strong baselines, and we will introduce a representative one, *i.e.*, TRADES (Zhang et al., 2019).

**TRADES** (Zhang et al., 2019) advocates a learning objective comprising two loss terms. Its first term penalizes the cross-entropy loss of benign training samples, and the second term regularizes the difference between benign output and the output of possibly malicious data points. Specifically, the worst-case Kullback-Leibler (KL) divergence between the output of each benign example and that of any suspicious data point in its $\epsilon$-bounded $l_\infty$ neighborhood is minimized in the regularization term:

$$\min_\Theta \sum_i (\mathrm{CE}(f(\mathbf{x}_i), y_i) + \beta \max_{\mathbf{x}' \in \mathcal{B}_\epsilon[\mathbf{x}_i]} \mathrm{KL}(f(\mathbf{x}'_i), f(\mathbf{x}_i))). \tag{3}$$

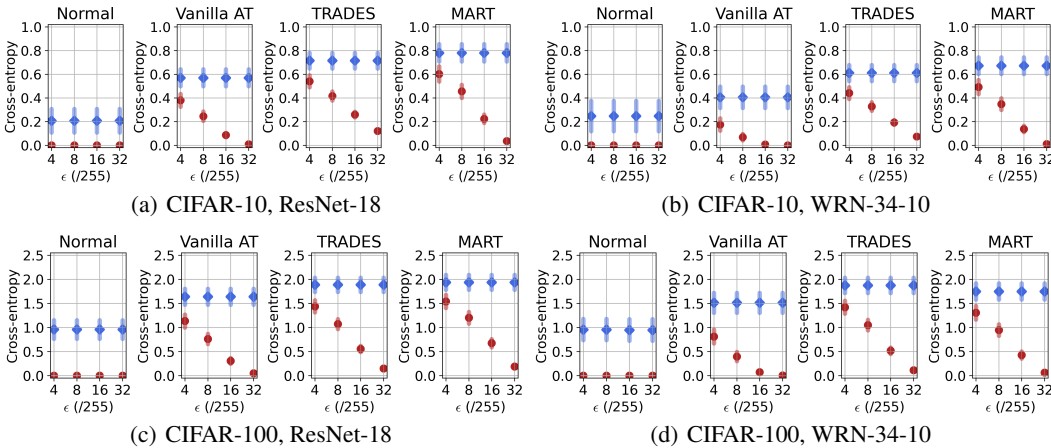

Figure 1: The average cross-entropy loss value of benign examples (blue) and collaborative examples (red) on (a) ResNet-18 trained using CIFAR-10, (b) wide ResNet-34-10 trained using CIFAR-10, (c) ResNet-18 trained on CIFAR-100, and (d) wide ResNet-34-10 trained on CIFAR-100. Shaded areas indicate scaled standard deviations. The collaborative examples are crafted with a fixed step size of $\alpha = 1/255$ and various perturbation budgets.

Other efforts have also been devoted in the family of adversarial training research, *e.g.*, MART (Wang et al., 2020), robust self training (RST) (Carmon et al., 2019), and adversarial weight perturbation (AWP) (Wu et al., 2020b). More specifically, after investigating the influence of mis-classified samples on model robustness, MART advocates giving specific focus to these samples for robustness. AWP identifies that flatter loss changing with respect to parameter perturbation leads to improved generalization of adversarial training, and provides a novel double perturbation mechanism. RST proposes to boost adversarial training by using unlabeled data and incorporating semi-supervised learning. Rebuffi et al. (2021) focus on data augmentation and study the performance of using generative models. There are also insightful papers that focuses on model architectures (Huang et al., 2021; Wu et al., 2020a; Bai et al., 2021; Mao et al., 2021; Paul & Chen, 2021), batch normalization (Xie et al., 2020a), and activation functions (Xie et al., 2020b; Dai et al., 2021). Distillation from robust models has also been studied (Zi et al., 2021; Shao et al., 2021; Awais et al., 2021).

Our ST for improving adversarial robustness is partially inspired by recent adversarial training effort, and we will discuss and compare to TRADES and other state-of-the-arts in Section 4.2, 5.1, and 5.2. Besides, our method can be naturally combined with a variety of other prior effort introduced in this section, to achieve further improvements, as will be demonstrated in Section 5.

## 3 COLLABORATIVE EXAMPLES

With the surge of interest in adversarial examples, we have achieved some understandings of plateau regions on the loss landscape of DNNs. However, valleys of the landscape seem less explored. In this section, we examine the valleys and explore the existence of collaborative examples, *i.e.*, data points are capable of achieving extremely lower classification loss, in the $\epsilon$-bounded neighborhood of benign examples. In particular, we discuss how adversarial robustness of DNNs and the collaborative examples affect each other, by providing several intriguing observations.

### 3.1 VALLEY OF THE LOSS LANDSCAPE

Unlike adversarial examples that are data points with higher or even maximal prediction loss, we pay attention to local minimum around benign examples in this subsection.

To achieve this, we here simply adapt the PGD method to instead *minimize* the prediction loss with:

$$\mathbf{x}^{t+1} = \mathbf{\Pi}_{\mathcal{B}_\epsilon[\mathbf{x}]}(\mathbf{x}^t - \alpha \cdot \text{sign}(\nabla_{\mathbf{x}^t}\text{CE}(f(\mathbf{x}^t), y))). \tag{4}$$

Comparing Eq. (4) to (1), it can be seen that their main difference is that, in Eq. (4), gradient descent is performed rather than gradient ascent. Similar to the I-FGSM and PGD attack, we clip the result in Eq. (4) after each update iteration to guarantee that the perturbation is within a presumed budget, *e.g.*, 4/255, 8/255, 16/255, or 32/255. We perform such an update with a step size of $\alpha = 1/255$. ResNet (He et al., 2016b) and wide ResNet (Zagoruyko & Komodakis, 2016) models trained on CIFAR-10 and CIFAR-100 are tested.

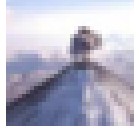 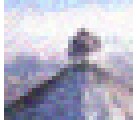 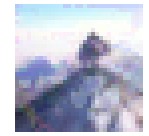

mountain     collaborative "mountain"   collaborative "mountain"
(normal)          (robust)

Figure 2: Visualization of benign example (left), collaborative example crafted on a normally trained ResNet-18 model (middle), and collaborative example crafted on a robust ResNet-18 model (right).

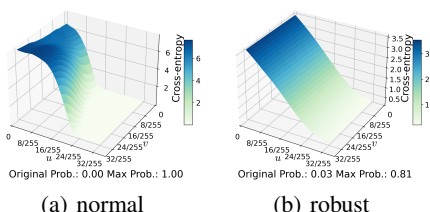

(a) normal         (b) robust

Figure 3: The loss landscapes in (a) and (b) show a normal ResNet-18 and an adversarially trained ResNet-18 using TRADES, respectively, all on CIFAR-100.

After multiple update steps (preciously 100 steps for this experiment), we evaluate the cross-entropy loss of the obtained examples. We compare it to the benign loss in the left most panels in Figure 1. It can be seen that, though benign data shows relatively low cross-entropy loss (*i.e.*, $\sim 0.2$ on average for ResNet-18 on the CIFAR-10 test set) already, there always exists neighboring data that easily achieve extremely lower loss values (*i.e.*, almost 0). Such data points showing lower prediction loss are collaborative examples of our interest. Figure 2 visualize the collaborative examples.

The existence of collaborative examples implies large local Lipschitz constants or non-flat regions of $g = \ell \circ f$, from a somehow different perspective against the conventional adversarial phenomenon. To shed more light on this, we further test with DNN models that were trained to be robust to adversarial examples, using the vanilla AT, TRADES, and MART [1]. The results can be found in the right panels in Figure 1. We see that it is more difficult to achieve zero prediction loss with these models, probably owing to flatter loss landscapes (Li et al., 2018). See Figure 3, in which the perturbation direction $u$ is obtained utilizing Eq. (4), and $v$ is random chosen in a hyperplane orthogonal to $u$. We analyze the angle between collaborative perturbations and PGD adversarial perturbations in Appendix B, and the results show that, on a less robust model, the collaborative perturbations and the adversarial perturbations are closer to be orthogonal.

Figure 2 also illustrate an collaborative example generated on the robust model, and we see that the collaborative perturbations for robust and non-robust models are perceptually different. In particular, a bluer sky leads to more confident prediction of the robust model. For readers who are interested, more visualization results are provided in Appendix F.

### 3.2 How Can Collaborative Examples Aid?

Given the results that the collaborative examples are less "destructive" on a more adversarially robust model, we raise a question:

*How can collaborative examples in return benefit adversarial robustness?*

Towards answering the question, one may first try to incorporate the collaborative examples into the training phase to see whether adversarial robustness of the obtained DNN model can be improved. To this end, we resort to the following learning objective:

$$\min_{\Theta} \sum_i (\text{CE}(f(\mathbf{x}_i), y_i) + \beta \cdot \text{KL}(f(\mathbf{x}_i^{col}), f(\mathbf{x}_i))), \tag{5}$$

where $\mathbf{x}_i^{col}$ is a collaborative example crafted using the method introduced in Section 3.1 and Eq. (4).

Such an optimization problem minimizes output discrepancy between the collaborative examples and their corresponding benign examples, in addition to the loss term that encourages correct prediction on benign examples. This simple and straightforward method has been similarly adopted in Tao *et al.*'s work (Tao et al., 2022) for resisting hypocritical examples. A quick experiment is performed here to test its benefit to adversarial robustness in our settings. The inner update for obtaining collaborative examples is performed over 10 steps, with a step size of $\alpha = 2/255$ and a perturbation budget of $\epsilon = 8/255$. We evaluate prediction accuracy on the PGD adversarial examples and benign examples for comparison. Figure 4 shows the test-set performance of ResNet-18 trained as normal and trained using Eq. (5) on CIFAR-10 and CIFAR-100, respectively. It can be seen that the robust accuracy is improved remarkably by solely incorporating collaborative examples into training. In the meanwhile, those normally trained models consistently show $\sim 0\%$ robust accuracy.

---

[1]All models trained via adversarial training show quite good robust accuracy under adversarial attacks on CIFAR-10 and CIFAR-100. Clean and robust accuracy of these models are provided in Appendix A.

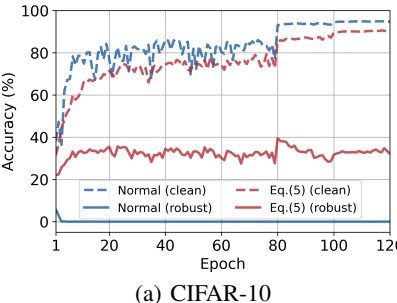 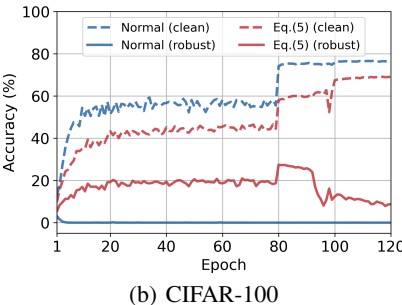

(a) CIFAR-10            (b) CIFAR-100

Figure 4: Changes in clean and robust accuracy during training ResNet-18 on (a) CIFAR-10 and (b) CIFAR-100, using Eq. (5). Best viewed in color.

However, there still exists a considerable gap between the obtained performance of Eq. (5) and that of TRADES (see Figure 9 in Appendix). Robust overfitting (Rice et al., 2020) can be observed on the red curves in Figure 4 (especially after the 80-th epoch), even though training with collaborative examples and testing with adversarial examples, and it seems more severe in comparison to that occurs with existing adversarial training methods. We also test other DNN architectures, *e.g.*, VGG (Simonyan & Zisserman, 2015) and wide ResNet (Zagoruyko & Komodakis, 2016), and the results are similar.

## 4 SQUEEZE TRAINING (ST)

In the above section, we have experimentally shown that collaborative examples exist and they can be used to improve the adversarial robustness of DNNs, by simply enforcing their output probabilities to be close to the output of their corresponding benign neighbors. In this section, we consider utilizing collaborative examples and adversarial examples jointly during model training, aiming at regularizing non-flat regions (including valleys and plateaus) of the loss landscape altogether.

### 4.1 METHOD

The adversarial examples can be utilized during training in a variety of ways, leading to various adversarial training methods. In this paper, considering that the adversarial and collaborative examples are both caused by non-flatness of the loss landscapes, we propose to penalize the maximum possible output discrepancy of any two data points within the $\epsilon$-bounded neighborhood of each benign example. Inspired by the adversarial regularization in, *e.g.*, TRADES, we adopt a benign prediction loss term in combination with a term in which possible adversarial examples and possible collaborative examples are jointly regularized. The benign example itself is not necessarily involved in error accumulation from the regularization term, since, in this regard, the output of a benign example is neither explicitly encouraged to be "adversarial" nor to be "collaborative". To achieve this, we advocate squeeze training (ST) whose learning objective is:

$$\min_{\Theta} \sum_i (\text{CE}(f(\mathbf{x}_i), y_i) + \beta \max_{\substack{\mathbf{x}' \in \mathcal{B}_\epsilon[\mathbf{x}_i] \\ \mathbf{x}'' \in \mathcal{B}_\epsilon[\mathbf{x}_i]}} \ell_{reg}(f(\mathbf{x}'_i), f(\mathbf{x}''_i)))$$

$$\text{s.t.} \quad p_f(y_i \,|\, \mathbf{x}'_i) \geq p_f(y_i \,|\, \mathbf{x}_i) \geq p_f(y_i \,|\, \mathbf{x}''_i), \; \forall i,$$

(6)

where $\ell_{reg}(\cdot)$ is a regularization function which evaluates the discrepancy between two probability vectors, and $\beta$ is a scaling factor which balances clean and robust accuracy. ST squeezes possible prediction gaps within the whole $\epsilon$-bounded neighborhood of each benign example, and it jointly regularizes the two sorts of non-flat regions of the loss landscape. The constraint in Eq. (6) is of the essence and is introduced to ensure that the two examples obtained in the inner optimization include one adversarial example and one collaborative example, considering it makes little sense to minimize the gap between two adversarial examples or between two collaborative examples. There are several different choices for the regularization function $\ell_{reg}$, *e.g.*, the Jensen–Shannon (JS) divergence, the squared $l_2$ distance, and the symmetric KL divergence, which are formulated as follows:

(1) JS:

$$\ell_{reg} = \frac{1}{2}(\text{KL}(\frac{f(\mathbf{x}') + f(\mathbf{x}'')}{2}, f(\mathbf{x}')) + \text{KL}(\frac{f(\mathbf{x}') + f(\mathbf{x}'')}{2}, f(\mathbf{x}''))),$$

(2) (Squared) $l_2$:

$$\ell_{reg} = \|f(\mathbf{x}') - f(\mathbf{x}'')\|_2^2,$$

---

**Algorithm 1** Squeeze Training (ST)

**Input:** A set of benign example and their labels $S$, number of training iterations $T$, learning rate $\eta$, number of inner optimization steps $K$, perturbation budget $\epsilon$, step size $\alpha$, and a choice of regularization function $\ell_{reg}$,
**Initialization:** Perform random initialization for $f$,
**for** $t = 1, \ldots, T$ **do**
    Sample a mini-batch of training data $\{(\mathbf{x}_i, y)\}_{i=1}^{m}$
    **for** $i = 1, \ldots, m$ (in parallel) **do**
        $\mathbf{x}_i' \leftarrow \mathbf{x}_i + 0.001 \cdot \mathcal{N}(\mathbf{0}, \mathbf{I})$, $\mathbf{x}_i'' \leftarrow \mathbf{x}_i + 0.001 \cdot \mathcal{N}(\mathbf{0}, \mathbf{I})$
        **while** $K \geq 0$ **do**
            $\mathbf{x}_i^{adv} \leftarrow \underset{\tilde{\mathbf{x}}_i \in \{\mathbf{x}_i, \mathbf{x}_i', \mathbf{x}_i''\}}{\arg\max} \mathrm{CE}(f(\tilde{\mathbf{x}}_i), y_i)$, $\mathbf{x}_i^{col} \leftarrow \underset{\tilde{\mathbf{x}}_i \in \{\mathbf{x}_i, \mathbf{x}_i', \mathbf{x}_i''\}}{\arg\min} \mathrm{CE}(f(\tilde{\mathbf{x}}_i), y_i)$
            $g_{inner} = \ell_{reg}(f(\mathbf{x}_i^{adv}), f(\mathbf{x}_i^{col}))$
            $\mathbf{x}_i' \leftarrow \mathbf{\Pi}_{\mathcal{B}_\epsilon[\mathbf{x}_i]}(\mathbf{x}_i^{adv} + \alpha \cdot \mathrm{sign}(\nabla_{\mathbf{x}_i^{adv}} g_{inner}))$, $\mathbf{x}_i'' \leftarrow \mathbf{\Pi}_{\mathcal{B}_\epsilon[\mathbf{x}_i]}(\mathbf{x}_i^{col} + \alpha \cdot \mathrm{sign}(\nabla_{\mathbf{x}_i^{col}} g_{inner}))$
            $K \leftarrow K - 1$
        **end while**
        $g_i \leftarrow \mathrm{CE}(f(\mathbf{x}_i), y_i) + \beta \cdot \ell_{reg}(f(\mathbf{x}_i^{adv}), f(\mathbf{x}_i^{col}))$
    **end for**
    $\Theta \leftarrow \Theta - \eta \frac{1}{m} \sum_{i=1}^{m} \nabla_\Theta g_i$
**end for**
**Output:** A robust classifier $f$ parameterized by $\Theta$.

---

(3) Symmetric KL:

$$\ell_{reg} = \frac{1}{2}(\mathrm{KL}(f(\mathbf{x}'), f(\mathbf{x}'')) + \mathrm{KL}(f(\mathbf{x}''), f(\mathbf{x}'))).$$

Among these choices, the (squared) $l_2$ and JS divergence are already symmetric, and we also adapt the original KL divergence to make it satisfy the symmetry axiom of desirable metrics, such that the adversarial examples and collaborative examples are treated equally.

With the formulation in Eq. (6), pairs of collaborative and adversarial examples are obtained simultaneously. The inner optimization are different from Eq. (1) and (4). The procedure is implemented as in Algorithm 1. At each training iteration, we perform $K$ update steps for the inner optimization, and at each step, the two examples are re-initialized by selecting from a triplet based on their cross-entropy loss and updated using sign of gradients. The comparison of cross-entropy loss for re-initialization from the triplet is carried out just to guarantee the chained inequality in Eq. (6).

## 4.2 DISCUSSIONS

Some discussions about ST are given. We would like to first mention that, although Eq. (6) is partially inspired by TRADES, the collaborative examples can generally and naturally be introduced to other adversarial training formulations.

Then, we compare ST (which optimizes Eq. (6)) to TRADES carefully. We know that the latter regularizes KL divergence between benign examples and their neighboring data points. A neighboring data point whose output shows maximal KL divergence from the benign output, can be an adversarial example or a collaborative example actually. In Figure 5, we demonstrate the ratio of collaborative examples and the prediction loss of different examples along with the training using TRADES proceeds. It can be seen that the ratio of collaborative examples used for model training decreases consistently, and being always less than $50\%$. Our ST aims to use $100\%$ of the collaborative and adversarial examples, if possible.

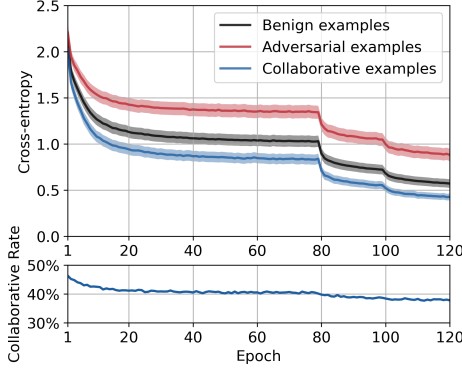

Figure 5: Changes in average cross-entropy loss of three types of examples along with TRADES training (upper), and changes in the ratio of collaborative examples it utilized (lower). The experiment is performed with ResNet-18 on CIFAR-10.

Comparing with TRADES, the $\ell_{reg}$ term in our ST imposes a stricter regularization, since the maximum possible output discrepancy between any two data points within the $\epsilon$-bounded neighborhood

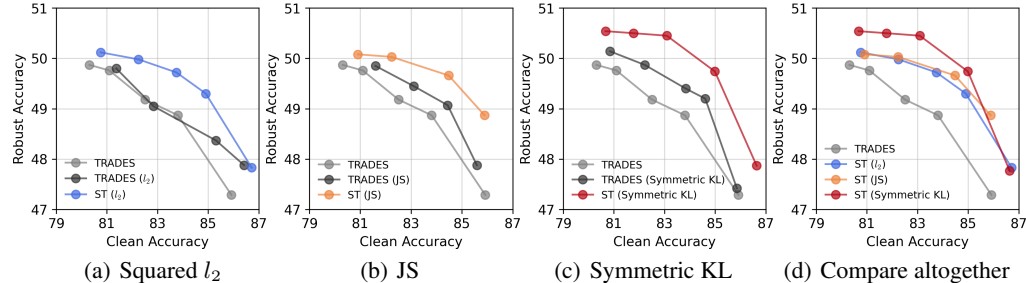

(a) Squared $l_2$      (b) JS      (c) Symmetric KL      (d) Compare altogether

Figure 6: Performance of TRADES and ST with different choices of the regularization functions $\ell_{reg}$ and the scaling factor $\beta$. The results are obtain on ResNet-18 and CIFAR-10. The robust accuracy is evaluated using AutoAttack with $\epsilon = 8/255$. For TRADES, we also try using the JS divergence, the squared $l_2$ distance, and the symmetric KL divergence for regularizing suspicious and benign outputs, as reported in (a), (b), and (c). Top right indicates better trade-off. Best viewed in color.

is penalized, which bounds the TRADES regularization from above. In this regard, the worst-case outputs (*i.e.*, adversarial outputs) and the best-case outputs (collaborative outputs) are expected to be optimized jointly and equally. Moreover, as has been mentioned, since ST does not explicitly enforce the benign output probabilities to match the output of adversarial examples, we expect improved trade-off between robust and clean errors. Extensive empirical comparison to TRADES in Section 5 will verify that our ST indeed outperforms it significantly, probably benefits from these merits.

Our ST adopts the same regularization loss for inner and outer optimization, and we also observed that, if not, moderate gradient masking occurs, *i.e.*, higher PGD-20 accuracy but lower classification accuracy under more powerful attacks, just like in Gowal et al. (2020)'s.

## 5 EXPERIMENTS

In this section, we compare the proposed ST to state-of-the-art methods. We mainly compare to vanilla AT (Madry et al., 2018), TRADES (Zhang et al., 2019), and MART (Wang et al., 2020). Performance of other outstanding methods (Zhang et al., 2020; Wu et al., 2020b; Kim et al., 2021; Yu et al., 2022) will be compared in Section 5.3, as additional data may be used. Experiments are conducted on popular benchmark datasets, including CIFAR-10, CIFAR-100 (Krizhevsky & Hinton, 2009), and SVHN (Netzer et al., 2011). Table 1 summarizes our main results, with ResNet (He et al., 2016b). We also test with *wide* ResNet (Zagoruyko & Komodakis, 2016) to show that our method works as well on large-scale classification models, whose results can be found in Table 3 and 4.

It is worth noting that our ST can be readily combined with many recent advances, *e.g.*, AWP (Wu et al., 2020b) and RST (Carmon et al., 2019). AWP utilizes weight perturbation in addition to input perturbation. We can combine ST with it by replacing its learning objective with ours. RST uses unlabeled data to boost the performance of adversarial training. It first produces pseudo labels for unlabeled data, and then minimizes a regularization loss on both labeled and unlabeled data.

**Training settings.** In most experiments in this section, we perform adversarial training with a perturbation budget of $\epsilon = 8/255$ and an inner step size $\alpha = 2/255$, except for the SVHN dataset, where we use $\alpha = 1/255$. In the training phase, we always use an SGD optimizer with a momentum of 0.9, a weight decay of 0.0005, and a batch size of 128. We train ResNet-18 (He et al., 2016a) for 120 epochs on CIFAR-10 and CIFAR-100, and we adopt an initial learning rate of 0.1 and cut it by $10\times$ at the 80-th and 100-th epoch. For SVHN, we train ResNet-18 for 80 epochs with an initial learning rate of 0.01, and we cut by $10\times$ at the 50-th and 65-th epoch. We adopt $\beta = 6$ for TRADES and $\beta = 5$ for MART by following their original papers. The final choice for the regularization function $\ell_{reg}$ and the scaling factor $\beta$ in our ST will be given in Section 5.1. All models are trained on an NVIDIA Tesla-V100 GPU.

**Evaluation details.** We evaluate the performance of adversarially trained models by computing their clean and robust accuracy. For robust accuracy, we perform various white-box attack methods including FGSM (Goodfellow et al., 2015), PGD (Madry et al., 2018), C&W's attack (Carlini & Wagner, 2017), and AutoAttack (AA) (Croce & Hein, 2020). Specifically, we perform PGD-20, PGD-100, and C&W$_\infty$ (*i.e.*, the $l_\infty$ version of C&W's loss optimized using PGD-100) under $\epsilon = 8/255$ and $\alpha = 2/255$. Since adversarial training generally shows overfitting (Rice et al., 2020), we select the model with the best PGD-20 performance from all checkpoints, as suggested in many recent papers (Zhang et al., 2020; Wu et al., 2020b; Wang et al., 2020; Gowal et al., 2021).

Table 1: Clean and robust accuracies of robust ResNet-18 models trained using different adversarial training methods. The robust accuracy is evaluated under an $l_\infty$ threat model with $\epsilon = 8/255$. We perform seven runs and report the average performance with $95\%$ confidence intervals.

| Dataset | Method | Clean | FGSM | PGD-20 | PGD-100 | C&W$_\infty$ | AA |
|---|---|---|---|---|---|---|---|
| CIFAR-10 | Vanilla AT | 82.78% ±0.12% | 56.94% ±0.17% | 51.30% ±0.16% | 50.88% ±0.26% | 49.72% ±0.24% | 47.63% ±0.08% |
| | TRADES | 82.41% ±0.12% | 58.47% ±0.19% | 52.76% ±0.08% | 52.47% ±0.13% | 50.43% ±0.17% | 49.37% ±0.08% |
| | MART | 80.70% ±0.17% | 58.91% ±0.24% | 54.02% ±0.29% | 53.58% ±0.30% | 49.35% ±0.27% | 47.49% ±0.23% |
| | ST (ours) | **83.10%** ±0.10% | **59.51%** ±0.22% | **54.62%** ±0.14% | **54.39%** ±0.16% | **51.43%** ±0.09% | **50.50%** ±0.07% |
| | TRADES+AWP | 81.16% ±0.12% | 57.86% ±0.14% | 54.56% ±0.06% | 54.45% ±0.14% | 50.95% ±0.12% | 50.31% ±0.10% |
| | ST (ours)+AWP | **82.53%** ±0.14% | **59.73%** ±0.14% | **55.56%** ±0.13% | **55.47%** ±0.13% | **52.05%** ±0.16% | **51.23%** ±0.12% |
| CIFAR-100 | Vanilla AT | 57.27% ±0.21% | 31.81% ±0.11% | 28.66% ±0.11% | 28.49% ±0.16% | 26.89% ±0.08% | 24.60% ±0.04% |
| | TRADES | 57.94% ±0.15% | 32.37% ±0.18% | 29.25% ±0.18% | 29.10% ±0.20% | 25.88% ±0.16% | 24.71% ±0.09% |
| | MART | 55.03% ±0.10% | 33.12% ±0.26% | 30.32% ±0.18% | 30.20% ±0.17% | 26.60% ±0.11% | 25.13% ±0.15% |
| | ST (ours) | **58.44%** ±0.12% | **33.35%** ±0.23% | **30.53%** ±0.13% | **30.39%** ±0.17% | **26.70%** ±0.20% | **25.61%** ±0.07% |
| | TRADES+AWP | 58.76% ±0.07% | 33.82% ±0.15% | 31.53% ±0.14% | 31.42% ±0.12% | 27.03% ±0.16% | 26.06% ±0.12% |
| | ST (ours)+AWP | **59.06%** ±0.08% | **34.50%** ±0.14% | **32.22%** ±0.14% | **32.16%** ±0.15% | **27.83%** ±0.11% | **26.86%** ±0.07% |
| SVHN | Vanilla AT | 89.21% ±0.27% | 59.81% ±0.29% | 51.18% ±0.29% | 50.35% ±0.27% | 48.39% ±0.18% | 45.96% ±0.21% |
| | TRADES | 90.20% ±0.20% | 66.40% ±0.18% | 54.49% ±0.13% | 54.18% ±0.15% | 52.09% ±0.10% | 49.51% ±0.16% |
| | MART | 88.70% ±0.20% | 64.16% ±0.24% | 54.70% ±0.26% | 54.13% ±0.29% | 46.95% ±0.24% | 44.98% ±0.17% |
| | ST (ours) | **90.68%** ±0.15% | **66.68%** ±0.22% | **56.35%** ±0.19% | **56.00%** ±0.22% | **52.57%** ±0.12% | **50.54%** ±0.10% |
| | TRADES+AWP | 89.80% ±0.21% | 66.30% ±0.19% | 59.01% ±0.20% | 58.63% ±0.22% | 54.72% ±0.11% | 52.54% ±0.12% |
| | ST (ours)+AWP | **90.77%** ±0.19% | **67.77%** ±0.25% | **59.95%** ±0.17% | **59.76%** ±0.20% | **55.26%** ±0.19% | **53.37%** ±0.19% |

## 5.1 COMPARISON OF DIFFERENT DISCREPANCY METRICS

To get started, we compare the three choices of discrepancy metric for $\ell_{reg}$, *e.g.*, the JS divergence, the squared $l_2$ distance, and the symmetric KL divergence. In Figure 6, we summarize the performance of ST with these choices. We vary the value of scaling factor $\beta$ to demonstrate the trade-off between clean and robust accuracy, and the robust accuracy is evaluated using AutoAttack (Croce & Hein, 2020) which provides reliable evaluations. For a fair comparison, we also evaluate TRADES with the original KL divergence function being replaced with these newly introduced discrepancy functions and illustrate the results in the same plots (*i.e.*, Figure 6(a), 6(b), and 6(c)) correspondingly. The performance curve of the original TRADES is shown in every sub-figure (in grey) in Figure 6. See Table 2 for all $\beta$ values in the figure. With the same $\ell_{reg}$, regularizations imposed by TRADES are less significant thus we use $\beta$ sets with larger elements.

From Figure 6(a) to 6(c), one can see that considerably improved trade-off between the clean and robust accuracy is achieved by using our ST, in comparison to TRADES using the same discrepancy metric for measuring the gap between probability vectors. Moreover, in Figure 6(d), it can be seen that our ST with different choices for the discrepancy functions always outperforms the original TRADES by a large margin, and using the symmetric KL divergence for $\ell_{reg}$ leads to the best performance overall. We will stick with the symmetric KL divergence for ST in our following comparison, and we use $\beta = 6$ for CIFAR-10, $\beta = 4$ for CIFAR-100, and $\beta = 8$ for SVHN.

Table 2: The scaling factor $\beta$ for different discrepancy functions reported in Figure 6.

| $\ell_{reg}$ | Method | $\beta$ set |
|---|---|---|
| KL | TRADES | {2, 4, 6, 8, 10} |
| $l_2$ | TRADES | {6, 8, 16, 24} |
| | ST (ours) | {2, 3, 4, 6, 8} |
| JS | TRADES | {16, 24, 32, 48} |
| | ST (ours) | {6, 8, 12, 16} |
| Symmetric KL | TRADES | {4, 6, 8, 12, 16} |
| | ST (ours) | {2, 4, 6, 8, 10} |

## 5.2 COMPARISON TO STATE-OF-THE-ARTS

Table 1 reports the performance of our adversarial training method ST and its competitors. Intensive results demonstrate that ST outperforms the vanilla AT (Madry et al., 2018), TRADES (Zhang et al., 2019), and MART (Wang et al., 2020) significantly, gaining consistently higher clean and robust accuracy on CIFAR-10, CIFAR-100, and SVHN. In other words, our ST significantly enhances

adversarial robustness with less degradation of clean accuracy, indicating better trade-off between clean and robust performance. Specifically, on CIFAR-10, the best prior method shows classification accuracy of 49.37% and 82.41% on the adversarial and clean test sets, respectively, while our ST with symmetric KL achieves 50.50% (**+1.13%**) and 83.10% (**+0.69%**). Combining with AWP (Wu et al., 2020b), we further gain an absolute improvement of **+0.92%** and **+1.37%** in robust and clean accuracy, respectively, compared to TRADES+AWP, on CIFAR-10. Similar observations can be made on CIFAR-100 and SVHN. Complexity analyses of our method is deferred to Appendix C, and training curves are given in Figure 9 to demonstrate less overfitting than that in Figure 4.

In addition to the experiments on ResNet-18, we also employ larger-scale DNNs, *i.e.*, wide ResNet (WRN) (Zagoruyko & Komodakis, 2016). We train robust WRN models, including robust WRN-34-5 and robust WRN-34-10 on CIFAR-10, and we report experimental results in Table 3. Obviously, the WRN models lead to higher clean and robust accuracy than that of the ResNet-18. Importantly, our ST still outperforms competitors on these networks, showing that the effectiveness of our method holds when the size of DNN model scales. Another WRN, *i.e.*, WRN-28-10, is also tested and the same observations can be made. We will test with it carefully in Section 5.3, in which additional unlabeled data is utilized during training.

Table 3: Evaluation using WRNs on CIFAR-10. It can be seen that the effectiveness of our method holds when the size of DNN scales. $PGD_{TRADES}$ is the default evaluation in (Zhang et al., 2019).

| Model | Method | Clean | $PGD_{TRADES}$ | AA |
|---|---|---|---|---|
| WRN-34-5 | TRADES | 83.11% | 55.78% | 51.33% |
| | ST (ours) | **83.14%** | **57.10%** | **52.21%** |
| WRN-34-10 | TRADES | 84.80% | 56.65% | 52.94% |
| | MART | 84.17% | - | 51.10% |
| | ST (ours) | **84.92%** | **57.73%** | **53.54%** |

## 5.3 ADVERSARIAL TRAINING WITH ADDITIONAL DATA

RST (Carmon et al., 2019) is a recent work that confirms unlabeled data could also be properly incorporated into training for enhancing adversarial robustness. Here we consider a simple and direct combination with it. Recall that, in the RST paper, it extracted 500K unlabeled data from 80 Million Tiny Images (Torralba et al., 2008). To utilize these unlabeled images, it generates pseudo labels for them, then performs adversarial training on a set including all CIFAR-10 training images and the originally unlabeled data. Our ST can easily be incorporated after obtaining the pseudo labels.

We implement RST and our combination with it (called ST-RST) by following settings in the original paper of RST. Table 4 report empirical results. We then compare the performance of ST-RST to recent work that utilizes the same set of unlabeled data, *i.e.*, GAIR-RST (Zhang et al., 2020), AWP-RST (Wu et al., 2020b), BAT-RST (Kim et al., 2021), and RWP-RST (Yu et al., 2022). Their results are collected from official implementations. For RWP-RST, which is in fact one of the best solutions in the same setting on RobustBench (Croce et al., 2020), it achieves robust accuracy of 60.36% by sacrificing clean accuracy, while our ST-RST gains **+0.39%** and **+1.53%** in robust and clean accuracy comparing to it.

Table 4: Training WRN-28-10 with additional unlabeled data. All methods in the table use the same unlabeled data from RST's official GitHub repository. $PGD_{RST}$ is the evaluation method in the RST paper.

| Method | Clean | $PGD_{RST}$ | AA |
|---|---|---|---|
| RST | 89.66% | 62.09% | 59.54% |
| BAT-RST | 89.61% | - | 59.54% |
| GAIR-RST | 89.36% | - | 59.64% |
| AWP-RST | 88.25% | - | 60.04% |
| RWP-RST | 88.87% | - | 60.36% |
| ST-RST (ours) | **90.40%** | **63.55%** | **60.75%** |

## 6 CONCLUSION

In this paper, we have studied the loss landscape of DNN models (robust or not) and specifically paid more attention to the valley region of the landscapes where collaborative examples widely exist. We have verified that collaborative examples can be utilized to benefit adversarial robustness. In particular, we have proposed ST, a squeeze training method, to take both adversarial examples and collaborative examples into accounts, *jointly and equally*, for regularizing the loss landscape during DNN training, forming a novel regularization regime. Extensive experiments have shown that our ST outperforms current state-of-the-arts across different benchmark datasets and network architectures, and it can be combined with recent advances (including RST and AWP) to gain further progress in improving adversarial robustness.

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

## A  PERFORMANCE OF MODELS IN FIGURE 1

Table 5: The clean and robust accuracy of models used in Figure 1, and the robust accuracy is evaluated by AutoAttack.

|  | CIFAR-10, ResNet-18 | | CIFAR-10, WRN-34-10 | | CIFAR-100, ResNet-18 | | CIFAR-100, WRN-34-10 | |
|---|---|---|---|---|---|---|---|---|
|  | Clean | AA | Clean | AA | Clean | AA | Clean | AA |
| Normal | 95.09% | 0.00% | 95.28% | 0.00% | 76.33% | 0.00% | 79.42% | 0.00% |
| Vanilla AT | 82.66% | 47.62% | 86.90% | 48.31% | 57.72% | 24.67% | 61.82% | 25.39% |
| TRADES | 82.51% | 49.18% | 84.80% | 52.94% | 57.99% | 24.58% | 57.10% | 26.76% |
| MART | 81.15% | 47.28% | 83.62% | 50.93% | 55.28% | 25.15% | 57.99% | 27.12% |

## B  COLLABORATIVE AND ADVERSARIAL DIRECTIONS

We analyze the angle between collaborative perturbations and PGD adversarial perturbations. We summarize the experiments in Figure 7 After a bunch of update steps, we observe that it lies in a limited range around $90°$, which is unsurprising in the high dimensional input space. However, for more robust models, we see that the angle deviate more from $90°$, indicating that powerful collaborative and adversarial perturbations become more correlated on the robust landscapes.

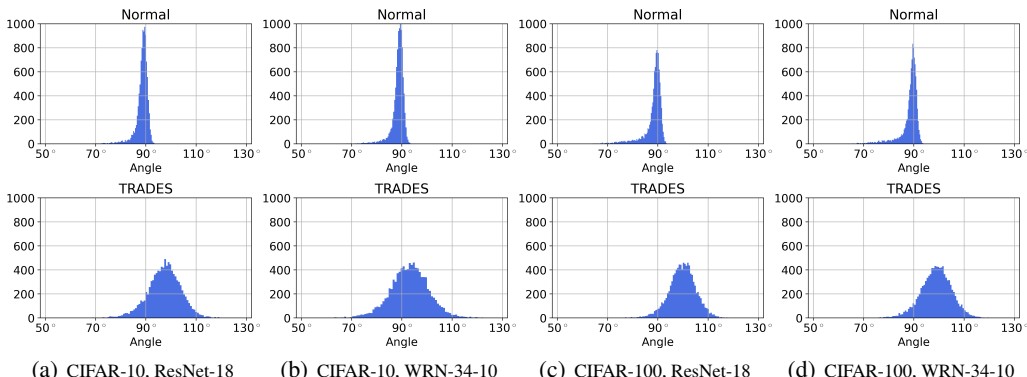

(a) CIFAR-10, ResNet-18   (b) CIFAR-10, WRN-34-10   (c) CIFAR-100, ResNet-18   (d) CIFAR-100, WRN-34-10

Figure 7: The distributions of angles between the PGD adversarial perturbations and collaborative perturbations. At the first update iteration, an collaborative example and its corresponding adversarial examples show opposite directions. However, after more and more update steps, since the gradient is computed w.r.t. different inputs, the adversarial and collaborative directions become less and less correlated and their angle finally lies in a range around $90°$. Interestingly, on a more robust model (TRADES), their correlation is more obvious and less concentrated around $90°$.

## C  COMPUTATIONAL COMPLEXITY OF ST

Since the adversarial examples and collaborative examples are both required in ST, the computational complexity in its inner optimization increases. Yet, we note that the two sorts of examples can be computed in parallel, thus the run time of our ST can be similar or only slightly higher than that of the baseline. In fact, even if we cut the number of inner iteration steps of our ST by $2\times$ to reduce its run time by roughly $2\times$, the performance of our method is still satisfactory. It shows robust accuracy (evaluated by AutoAttack) of $49.52\%$ and clean accuracy of $84.37\%$, which already surpasses the performance of Vanilla AT, TRADES, and MART.

Furthermore, the performance of previous state-of-the-arts does not always improve with higher computational capacity (*e.g.*, more inner optimization steps). For instance, TRADES show slightly better robust accuracy (AA: $49.76\% \pm 0.09\%$) but decreased clean accuracy ($81.57\% \pm 0.14\%$) on CIFAR-10 with $2\times$ more inner steps, which are both not better than our ST (AA: $50.50\% \pm 0.07\%$, and clean accuracy: $83.10\% \pm 0.10\%$).

We also consider generating two neighboring examples for each benign example and compute the mean/max of the regularization loss for out optimization for existing state-of-the-arts. Experimental results show that the mean of regularization is more effective, and Table 6 shows the ResNet results of all compared methods on CIFAR-10, CIFAR-100, and SVHN, with such an innovative formulation. We can see that none of the methods (including the vanilla AT[†], TRADES[†], and MART[†], where the symbol "[†]" indicates the utilization of two examples) improves consistently, comparing to the results in Table 1. Among all these competitors of ST, TRADES[†] performs the best. Table 6 demonstrates that its performance advances on CIFAR-10 and CIFAR-100 but drops on SVHN in comparison to the TRADES results in Table 1. With WRN architectures, it improves on WRN-28-10 (clean: $+0.34\%$, AA: $+0.42\%$) with additional unlabeled data, while drops on WRN-34-5 (clean: $-0.02\%$, AA: $-0.12\%$) and WRN-34-10 (clean: $-0.71\%$, AA: $-1.18\%$) without.

In fact, such an innovative formulation involving two examples is more likely to incorporate collaborative examples into TRADES[†], and this might be the reason why it works better with TRADES[†] than with the vanilla AT[†] and MART[†]. After all, combined with AWP or not, such an innovative formulation involving two examples is still obviously inferior to our ST.

Table 6: Two neighboring examples are generated for each benign example in the inner optimization of the vanilla AT[†], TRADES[†], and MART[†], and the mean of regularization is adopted for the outer optimization. The robust accuracy is still evaluated under an $l_\infty$ threat model with $\epsilon = 8/255$. We perform seven runs and report the average performance with $95\%$ confidence intervals.

| Dataset | Method | Clean | FGSM | PGD-20 | PGD-100 | C&W$_\infty$ | AA |
|---|---|---|---|---|---|---|---|
| CIFAR-10 | Vanilla AT[†] | 82.72% ±0.21% | 57.26% ±0.32% | 51.51% ±0.20% | 50.93% ±0.52% | 49.91% ±0.26% | 47.56% ±0.15% |
| | TRADES[†] | 82.59% ±0.07% | 58.77% ±0.31% | 53.25% ±0.20% | 52.96% ±0.18% | 51.03% ±0.29% | 49.47% ±0.09% |
| | MART[†] | 80.27% ±0.13% | 58.86% ±0.23% | 54.36% ±0.27% | 54.16% ±0.31% | 49.48% ±0.18% | 47.61% ±0.09% |
| | ST (ours) | **83.10%** ±0.10% | **59.51%** ±0.22% | **54.62%** ±0.14% | **54.39%** ±0.16% | **51.43%** ±0.09% | **50.50%** ±0.07% |
| | TRADES+AWP[†] | 81.43% ±0.07% | 58.59% ±0.14% | 55.13% ±0.11% | 54.99% ±0.12% | 51.64% ±0.12% | 50.50% ±0.07% |
| | ST (ours)+AWP | **82.53%** ±0.14% | **59.73%** ±0.14% | **55.56%** ±0.13% | **55.47%** ±0.13% | **52.05%** ±0.16% | **51.23%** ±0.12% |
| CIFAR-100 | Vanilla AT[†] | 56.57% ±0.11% | 31.71% ±0.20% | 28.52% ±0.19% | 28.27% ±0.21% | 27.11% ±0.18% | 24.64% ±0.12% |
| | TRADES[†] | 57.99% ±0.18% | 32.18% ±0.10% | 29.99% ±0.24% | 29.42% ±0.16% | 25.90% ±0.31% | 24.77% ±0.17% |
| | MART[†] | 55.06% ±0.09% | 32.40% ±0.22% | 30.52% ±0.17% | 30.17% ±0.14% | 26.42% ±0.17% | 24.92% ±0.09% |
| | ST (ours) | **58.44%** ±0.12% | **33.35%** ±0.23% | **30.53%** ±0.13% | **30.39%** ±0.17% | **26.70%** ±0.20% | **25.61%** ±0.07% |
| | TRADES+AWP[†] | 58.86% ±0.12% | 34.11% ±0.17% | 31.55% ±0.15% | 31.64% ±0.30% | 27.07% ±0.15% | 26.17% ±0.15% |
| | ST (ours)+AWP | **59.06%** ±0.08% | **34.50%** ±0.14% | **32.22%** ±0.14% | **32.16%** ±0.15% | **27.83%** ±0.11% | **26.86%** ±0.07% |
| SVHN | Vanilla AT[†] | 87.39% ±0.15% | 58.66% ±0.17% | 50.35% ±0.26% | 49.44% ±0.29% | 46.45% ±0.31% | 44.26% ±0.12% |
| | TRADES[†] | 88.16% ±0.13% | 63.23% ±0.25% | 55.06% ±0.20% | 54.54% ±0.26% | 51.00% ±0.24% | 48.81% ±0.11% |
| | MART[†] | 86.72% ±0.17% | 62.23% ±0.19% | 55.08% ±0.21% | 54.43% ±0.25% | 46.77% ±0.18% | 44.78% ±0.14% |
| | ST (ours) | **90.68%** ±0.15% | **66.68%** ±0.22% | **56.35%** ±0.19% | **56.00%** ±0.22% | **52.57%** ±0.12% | **50.54%** ±0.10% |
| | TRADES+AWP[†] | 90.32% ±0.20% | 67.55% ±0.27% | 58.67% ±0.27% | 58.35% ±0.24% | 54.72% ±0.27% | 52.67% ±0.10% |
| | ST (ours)+AWP | **90.77%** ±0.19% | **67.77%** ±0.25% | **59.95%** ±0.17% | **59.76%** ±0.20% | **55.26%** ±0.19% | **53.37%** ±0.19% |

# D  TRAINING CURVES OF ST

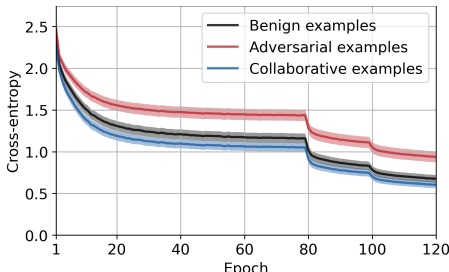

Figure 8: How the average cross-entropy loss changes during training using *our ST*. The experiment is performed with ResNet-18 on CIFAR-10. Shaded areas represent scaled standard deviations. It can be seen that the gap between the collaborative examples and the benign examples and that between the adversarial examples and the benign examples are both obviously reduced, comparing to Figure 5. Best viewed in color.

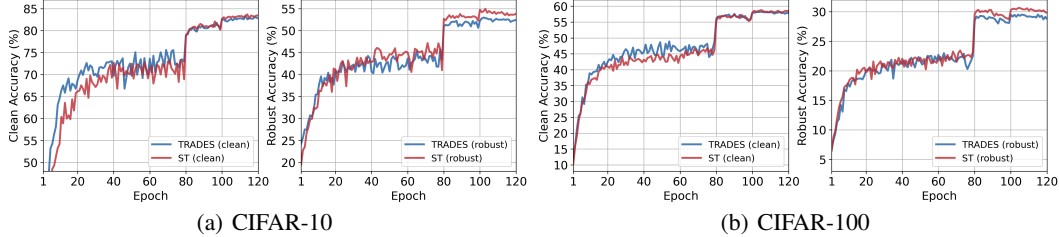

(a) CIFAR-10                                                      (b) CIFAR-100

Figure 9: Changes in clean and robust accuracy during training ResNet-18 on (a) CIFAR-10 and (b) CIFAR-100, using TRADES and *our ST*. The robust accuracy is evaluated using PGD adversarial examples generated over 20 steps with $\epsilon = 8/255$ and a step size of $\alpha = 2/255$. Best viewed in color.

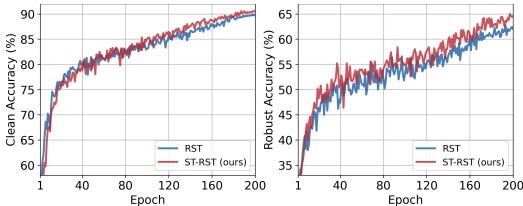

Figure 10: Changes in clean and robust accuracy during training WRN-28-10 on CIFAR-10, using RST and ST-RST. By following the original paper of RST, the robust accuracy is evaluated using PGD adversarial examples generated over 20 steps with $\epsilon = 0.031$ and a step size of $\alpha = 0.007$ on the first 500 images in the test set. Although we only train 200 epochs *as suggested in the RST paper* for comparison in Table 4 in our main paper, it can be seen that more training epochs can still be beneficial to our method. Best viewed in color.

# E  TO MAKE "ADVERSARIAL" AND "COLLABORATIVE" CLEAR

In this paper and especially Section 4, we sometimes generalize the definition of adversarial examples to include all data points (in the bounded neighborhood of benign examples) showing considerably higher prediction loss than that of the benign loss, in contrast to the definition in a narrower sense saying that different label prediction ought to be made. The collaborative examples are similarly "defined", somewhat non-rigorously. Likewise, throughout the paper, we abuse the word "*adversarial*" and "*collaborative*" to describe the spirit of achieving higher and lower loss than the benign loss, respectively.

# F    VISUALIZATIONS

Here we visualize some collaborative examples and collaborative perturbations in CIFAR-10, CIFAR-100, and SVHN.

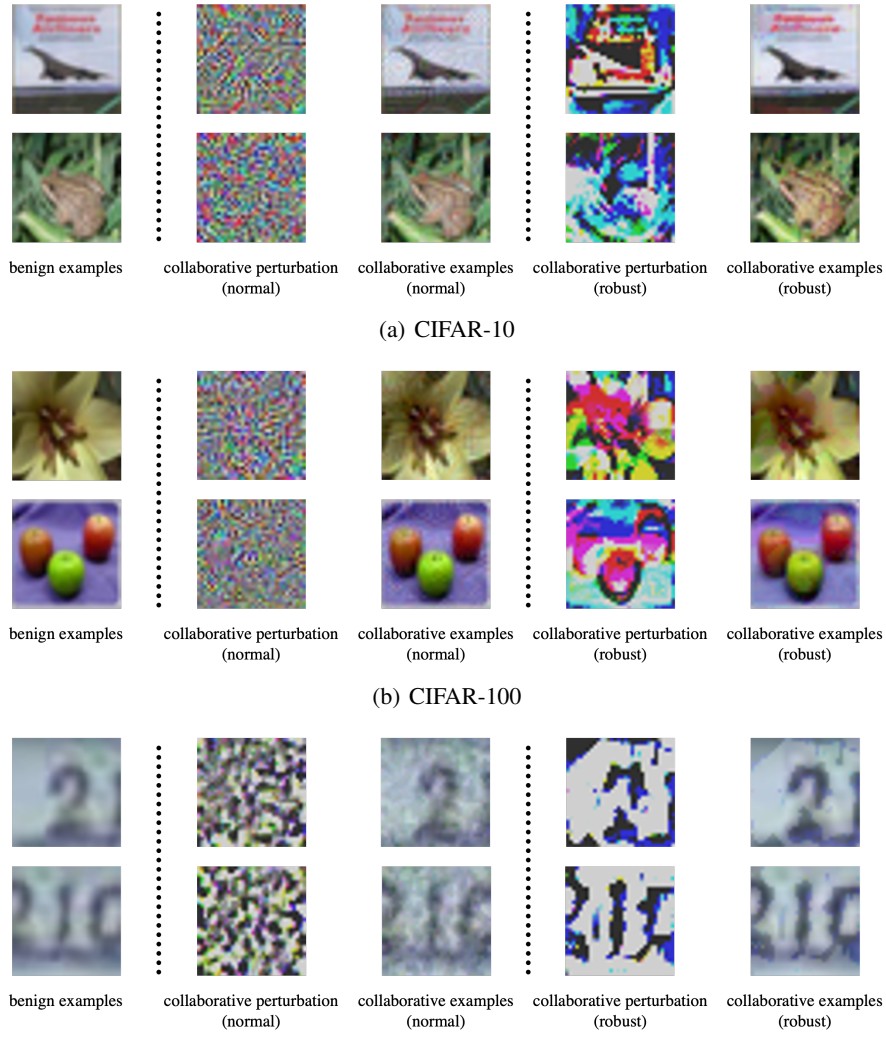

(a) CIFAR-10

(b) CIFAR-100

(c) CIFAR-100

Figure 11: Visualization of benign examples, collaborative perturbations, and collaborative examples in (a) CIFAR-10, (b) CIFAR-100, and (c) SVHN. The collaborative examples are crafted on normally trained ResNet-18 models and robust ResNet-18 models. Apparently, collaborative examples on robust and non-robust models are strikingly different.

