# OpenReview forum: "Squeeze Training for Adversarial Robustness"
_ICLR.cc/2023/Conference — ICLR 2023 poster_

### Official Review · Reviewer_46B8 · 2022-10-24

**Confidence:** 4
**Correctness:** 4
**Technical Novelty And Significance:** 3
**Empirical Novelty And Significance:** Not applicable
**Recommendation:** 6

**Clarity, Quality, Novelty And Reproducibility:**

As mentioned above the paper is easy to follow. The paper on its own is not very novel as the others are inspired by multiple other works. However, combining those ideas in adversarial training is novel indeed. Finally, I think the authors need to provide more details to make the results reproducible.

**Strength And Weaknesses:**

Strengths
1. The paper is well written. The main idea of the paper is presented in an effective manner. This is followed by initial results with just collaborative examples, then squeeze training and others.
2. The idea of using collaborative examples is an interesting one. The authors show that such examples alongside adversarial examples, not only improves robustness but also shows less degradation on the clean dataset. The results seem to hold across a large variety of datasets as well.
3. Another strength of the proposed method is its extendability to other adversarial training methods. The authors show that this can be done easily and that it still yields better results.

Weakness
1. While I like the idea of collaborative examples presented in this paper, the paper can still benefit from an explanation regarding why such examples would be useful in adversarial training. From the perspective of the model these examples are "easy". So, in principle the model should not learn many useful information from such examples. However, we see that when used in conjunction with adversarial examples and the regularization function the results are impressive.
2. In this work, the authors use the whole dataset when doing collaborative training. This should incur increased cost. I wonder if the whole dataset is necessary or not. Some analysis on this regard will also help shed further light to the nature of the collaborative examples.
3. Extending on the previous point, I would have liked to see a further analysis on the cost of doing squeeze training. I noticed some analysis in the appendix. However, I would have liked to seen a more in depth analysis regarding cost vs gain.
4. Similarly I would have liked to see more details about the experiments. I might have missed it but I did not see a lot of discussion about the split. In particular I want to know how the adversarial examples for the test cases were sampled.
5. The authors can also consider adding imagenet dataset in their experiments.


**Summary Of The Paper:**

In this paper, the authors tackle adversarial training. They pay more attention to the valley region of the loss landscape of DNNs. This is where collaborative examples exists. Collaborative examples fall within a bounded region of a benign example while having very small loss values. The authors first show that such examples can be directly used to improve the adversarial robustness of DNNs. Next they show that such examples in conjunction with adversarial examples (where they are both treated equally) can give an even higher boost to the robustness of DNNs. This is known as squeeze training as it "squeezes possible prediction gaps and jointly regularizes the two sorts of non-smooth and steep regions in the loss landscape". The authors compare their proposed method against multiple state of the art adversarial training methods. They also show how the choice of the regularization function impacts the robustness. Furthermore, they show that proposed method can also be extended to other adversarial training methods like RST.

**Summary Of The Review:**

In this empirical paper, the proposed squeeze method shows impressive performance across multiple datasets. However, as mentioned above some further analysis could elevate this paper further.

---

> ### Author Response · Authors · 2022-11-16
> **Response to Reviewer 46B8**
>
> Thanks for the positive feedback. Our response to the comments are provided as follows.
>
> > The paper can still benefit from an explanation regarding why such examples would be useful in adversarial training.
>
> Though not directly imposing restriction on adversarial robustness, regularizing the best-case prediction in fact optimizes a lower bound of the local Lipschitz constant, and the local Lipschitz constant is known to be related to adversarial robustness and landscape flatness. By incorporating collaborative examples and adversarial examples jointly and equally into training, ST attempts to obtain loss landscapes that are flat not only at the benign data points, but also at any data points within the $\epsilon$-bounded neighborhood of the benign data points. The lower bound can then be tighter.
>
> In addition, introducing the collaborative examples as in ST can release the benign samples from being penalized to match the output of adversarial examples. We conjecture that less conflict is then imposed against predicting benign samples, and a better trade-off between clean and robust accuracy can thus be achieved.
>
> Also, we would like to consider the collaborative examples as the ones that fool models to give **(abnormally) overconfident** predictions rather than easy examples [3]. We believe these examples in fact also show some vulnerability of the models.
>
> > I wonder if the whole dataset is necessary or not. Some analysis on this regard will also help shed further light to the nature of the collaborative examples.
>
> Simply reducing the number of training samples for performing adversarial training or collaborative training leads to slightly worse performance. Specifically, we tried using only 50% of the sample in each training batch for ST regularization, and achieved robust accuracy of 81.64% and clean accuracy of 49.18% on CIFAR-10. Another way of reducing the cost of ST is to reduce the maximal update step in its inner optimization, we can achieve robust accuracy of 84.37% and clean accuracy of 49.52% in this way, which already outperforms all other compared methods with 2x less runtime (which will also be much faster than them).
>
> > Analysis regarding cost vs gain.
>
> As has been discussed in Appendix C, computational complexity of our method is in theory larger than that of TRADES, yet 1) further increasing the inner steps of TRADES by 2x does not lead to much performance gain, 2) run time of our ST can be similar or only slightly higher than that of the baseline, with parallel computing, 3) as shown in our response to the previous comment, it is possible to cut the run time of our ST by 2x to achieve robust accuracy of 84.37% and clean accuracy of 49.52%, which is already superior to the vanilla AT, TRADES, and MART.
>
> We also tried to improve the capability of TRADES by introducing two adversarial examples for each benign sample in the training batch and regularizing the average/maximal of their prediction gap from the benign sample, following the suggestion of Reviewer 84B3. Experimental results still show that, with such an innovative formulation where the complexity of different methods are all equal, our ST still outperforms its competitors considerably. All related results have been included in Appendix C in the revised paper.
>
> > How the adversarial examples for the test cases were sampled.
>
> Just like most prior work, adversarial training for all methods were performed on the official training split of CIFAR-10 [1], CIFAR-100 [1], and SVHN [2], while test was performed on the official test set. That is, for CIFAR-10 and CIFAR-100, 50000 samples were used for training, and the test accuracy was calculated on the remaining 10000 samples. While for SVHN, we have 73257 training samples and 26032 test samples. For each test sample in these datasets, adversarial examples were generated based on FGSM, PGD-20, PGD-100, C&W, and AutoAttack, and the prediction accuracy of adversarial examples generated using each of these attacks is reported and compared.
>
> > The authors can also consider adding imagenet dataset in their experiments.
>
> With limited time for preparing rebuttal, it is super challenging to accomplish experiments on the full ImageNet dataset. In order to address the concern of the reviewer, we conducted an experiment on Tiny-ImageNet. Experimental results can be seen in the below table, and it shows that the performance gain of our ST is consistent also on ImageNet-like data.
>
> | Method | Clean Acc. | Robust Acc. (AA) |
> | :----: | :----: | :----: |
> | AT | 33.44% | 11.02% |
> | TRADES | 38.69% | 10.82% |
> | MART | 29.84% | 11.31% |
> | ST (ours) | **39.52%** | **11.58%** |
> | TRADES+AWP | 45.24% | 14.60% |
> | ST (ours)+AWP | **45.80%** | **15.42%** |
>
> &nbsp;
> &nbsp;
> &nbsp;
>
> [1] https://www.cs.toronto.edu/~kriz/cifar.html
> [2] http://ufldl.stanford.edu/housenumbers/
> [3] Müller R, Kornblith S, Hinton G E. When does label smoothing help?. NeurIPS, 2019.

---

> > ### Comment · Reviewer_46B8 · 2022-12-05
> > **Response to Author Feedback**
> >
> > I thank the authors for their responses. After going through the response and the other discussions in this thread I have decided not to update my scores at this point.

---

### Official Review · Reviewer_Q4iB · 2022-10-25

**Confidence:** 4
**Correctness:** 3
**Technical Novelty And Significance:** 3
**Empirical Novelty And Significance:** 4
**Recommendation:** 6

**Clarity, Quality, Novelty And Reproducibility:**

The paper is clear, well-written and has sufficient novelty. Hyperparameters used have been discussed. The authors are encouraged to share the code as well.

**Strength And Weaknesses:**


Strengths -
- This work builds upon existing methods to propose a simple and intuitive change that shows consistent performance gains.
- The observation that merely using collaborative examples can improve adversarial robustness is interesting.
- The observation that TRADES training indeed results in the generation of collaborative examples in some cases is surprising and interesting.
- The paper is well written and clear.

Weaknesses -
- The performance gains are marginal - hence it is important to perform more ablation experiments to ensure that the gains are indeed due to the use of collaborative examples. (some suggestions mentioned below)
- Attack generation part of Algorithm-1 can be discussed in more detail. Specifically, how does this attack ensure max and min loss, rather than two examples which are far from each other?

Questions for rebuttal -
- It is possible that collaborative example based training causes gradient masking and hence shows improved robustness. Could the authors share AutoAttack accuracy of the best epoch of each case in Fig.4?
- Could the authors also compare with TRADES defense where CE loss is used in the inner maximization step, as discussed in [1]?
- It is not clear how well the inner maximization can generate collaborative and adversarial examples given a common generation step. Could the authors compare the loss of the resulting examples with the loss obtained when CE loss is maximized or minimized independently?
- It would be interesting to see what happens when adversarial examples are first generated by maximizing CE loss, and in a next step, collaborative examples are generated to maximize the KL divergence w.r.t. the adversarial examples.

[1] Gowal et al., Uncovering the Limits of Adversarial Training against Norm-Bounded Adversarial Examples


**Summary Of The Paper:**

The authors propose a simple modification to existing adversarial training methods - rather than merely using adversarial examples that are meant to maximize classification loss, they propose to use collaborative examples which minimize the training loss. The training formulation minimizes a distance such as the symmetric KL divergence between the adversarial and collaborative examples to impose a stricter smoothness constraint when compared to existing methods. The method shows consistent gains over existing methods on CIFAR-10, CIFAR-100 and SVHN, and scales to WideResNet models as well.

**Summary Of The Review:**

The work presents a simple and intuitive change to existing adversarial training algorithms and shows improvements over existing methods. Some more ablations would ascertain the importance of collaborative examples further. Hence I recommend a borderline accept for now.

--- Post Rebuttal update ---

I thank the authors for the rebuttal and I would like to maintain my score.

---

> ### Author Response · Authors · 2022-11-16
> **Response to Reviewer Q4iB (part 1/2)**
>
> Thanks for your positive feedback. Our response to the comments are provided as follows.
>
> > The performance gains are marginal.
>
> We would like to politely mention that our ST managed to outperform state-of-the-arts on CIFAR-10, CIFAR-100, and SVHN consistently by a considerable margin, e.g., on CIFAR-10, it achieves an absolute accuracy gain of >1.0% und AutoAttack which is super challenging [3]. When additional unlabeled data is introduced, e.g., 500K unlabeled data from 80 Million Tiny Images, our ST outperforms the performance of all previous state-of-the-arts on RobustBench in the same setting (without modifying the model architecture as in [1] or introducing synthesized high quality images as in [4]) considerably. All these results demonstrate that the performance gain of our method is not marginal.
>
> > Attack generation part of Algorithm 1 can be discussed in more detail. Specifically, how does this attack ensure max and min loss, rather than two examples which are far from each other?
>
> Thanks for the suggestion. We have revised Section 4 accordingly to avoid possible misunderstanding. Our ST aims to penalize the maximum possible output discrepancy of any two data points within the $\epsilon$-bounded neighborhood of each benign training sample, rather than to bridge the gap between the data points with highest and lowest prediction loss. At each training iteration, a pair of examples should be generated, for which $K$ update steps are performed. We expect the pair of examples include a collaborative example which shows lower prediction loss than the benign one and an adversarial example which shows higher prediction, thus a chained equality constraint is introduced in Eq. (6). To guarantee such a constraint, we initialize the two examples by selecting from a triplet and updating using sign of gradients at each of the $K$ steps.
>
> Specifically, in ST, the same regularization loss is used for the inner maximization problem and the outer minimization problem, such that the comparison between ST and TRADES is fair. In fact, if adversarial examples and collaborative examples are generated separately in the inner optimization, gradient masking can be observed, just like in [1] (where gradient masking occurs when the margin loss is adopted for inner optimization though), i.e., higher PGD accuracy but much lower AutoAttack accuracy.
>
> > It is possible that collaborative example based training causes gradient masking and hence shows improved robustness? Could the authors share AutoAttack accuracy of the best epoch of each case in Fig.4?
>
> Following the suggestion, we have evaluated the AutoAttack accuracy of the best epoch of each case in Figure 4. The best epoch of Eq. (5) obtains 35.61% and 22.68% on CIFAR-10 and CIFAR-100, respectively, while normal training achieves 0% accuracy.
>
> For Table 1, 3, and 4 which compare our ST to previous SOTA systematically, the AutoAttack accuracy has already been reported for all compared methods, in addition to FGSM, PGD-20, PGD-100, and C&W results. We can see that the gain of our ST is consistent under all these attacks, indicating that its superiority does NOT come from gradient masking against weak attacks [1-2].
>
> > Could the authors also compare with TRADES defense where CE loss is used in the inner maximization step, as discussed in [1]?
>
> We performed the suggested experiment by training ResNet-18 on CIFAR-10. With $\beta=6$, TRADES+CE achieves clean accuracy of 81.48% and robust accuracy of 49.67%, which are still obviously inferior to our ST results. Detailed results with various $\beta$ values are summarized as follows. Note that multiple runs have been performed and the average performance is reported. With $\beta=6$, slightly improved robust accuracy can be achieved using TRADES+CE (compared to the original TRADES which shows 49.37%), yet the clean accuracy is lower (81.48% vs 82.41%), unlike the results on WRN-28-10 reported in [1]. We can also compare the robust accuracy on WRN-28-10, for which our ST+RST obtains 60.75% while TRADES+CE+RST achieves 59.45% [1], both leveraging 80M Tiny Images.
>
> |            | TRADES+CE  |            | ST         |             |
> |------------|:------------:|:-------------:|:------------:|:-------------:|
> |            | Clean Acc. | Robust Acc. (AA) | Clean Acc. | Robust Acc. (AA) |
> | $\beta=2$  |     84.76%     |   48.14%      |   86.64%   |      47.77%    |
> | $\beta=4$  |      82.72%     |    49.01%      |     84.99%       |    49.74%       |
> | $\beta=6$  |      81.48%      |      49.67%     |      83.10%    |       50.50%      |
> | $\beta=8$  |     80.08%       |       49.53%      |     81.77%       |     50.52%        |
> | $\beta=10$  |     78.46%       |       49.08%      |     80.67%       |     50.54%        |

---

> > ### Author Response · Authors · 2022-11-16
> > **Response to Reviewer Q4iB (part 2/2)**
> >
> > > It is not clear how well the inner maximization can generate collaborative and adversarial examples given a common generation step. Could the authors compare the loss of the resulting examples with the loss obtained when CE loss is maximized or minimized independently?
> >
> > As has been explained in our response to the second comment, ST focuses on penalizing the maximum possible output discrepancy of any two data points. Although the constraint in Eq. (6) encourages that the pair of generated examples include an adversarial example which shows higher prediction loss (than the benign loss) and a collaborative example which shows lower prediction loss, they do not necessarily achieve the maximal and minimal loss in the $\epsilon$-bounded neighborhood of a benign sample. Section 4.1 has been revised to make it clearer.
> >
> > In fact, since ST regularizes the flatness of loss landscape, the maximal possible prediction loss and minimal possible prediction loss in the neighborhood are also regularized. On ResNet-18, ST achieves 1.29 and 0.53, while normal training achieves 24.92 and 0 for the maximal and minimal loss.
> >
> > > It would be interesting to see what happens when adversarial examples are first generated by maximizing CE loss, and in a next step, collaborative examples are generated to maximize the KL divergence w.r.t. the adversarial examples.
> >
> > We performed the experiments as suggested, and the resulting model achieves clean accuracy of 81.74% and robust accuracy of 49.82% (under AutoAttack), which is no better than our ST (83.10% clean accuracy and 50.50% robust accuracy), indicating that it is indeed more effective to regularize the valley and plateau of landscapes **jointly** and **equally**, as in our ST.
> >
> > &nbsp;
> > &nbsp;
> > &nbsp;
> >
> > [1] Gowal S, Qin C, Uesato J, et al. Uncovering the limits of adversarial training against norm-bounded adversarial examples. arXiv preprint arXiv:2010.03593, 2020.
> > [2] Athalye A, Carlini N, Wagner D. Obfuscated gradients give a false sense of security: Circumventing defenses to adversarial examples. ICML 2018.
> > [3] Croce F, Andriushchenko M, Sehwag V, et al. Robustbench: a standardized adversarial robustness benchmark. arXiv preprint arXiv:2010.09670, 2020.
> > [4] Gowal S, Rebuffi S A, Wiles O, et al. Improving robustness using generated data. NeurIPS 2021.

---

### Official Review · Reviewer_84B3 · 2022-10-26

**Confidence:** 4
**Correctness:** 4
**Technical Novelty And Significance:** 3
**Empirical Novelty And Significance:** 3
**Recommendation:** 8

**Clarity, Quality, Novelty And Reproducibility:**

Clarity: The paper is easy to understand.

Quality: The experimental results are compelling but some additional analysis/justification of the results would be beneficial.

Novelty: The core idea is novel to the best of my knowledge.

Reproducibility: The paper explain the high level ideas needed to reproduce their work but do not provide code at this point.

**Strength And Weaknesses:**

Strenghts:

- The idea of minimizing the maximum prediction difference across arbitrary pairs of points within the Lp ball is interesting and, to the best of my knowledge, novel.
- The experimental results do show a consistent improvement in the robustness and accuracy of the resulting model, even against strong attacks such as Auto-Attack.

Weaknesses:

- I did not find the discussion around “collaborative examples” particularly insightful. The existence of such examples is more or less implied by the existence of adversarial examples in the first place (if increasing the loss with small perturbations is easy, one would expect that decreasing the loss would be easy as well).
- The motivation of the proposed method is not entirely clear. If the goal is to reduce the worst-case error, why does the best-case error have an effect? Is there a fundamental reason or does it happen to help make the loss smoother in practice?
- Relatedly, the proposed method cannot be directly compared with existing approaches since (as the authors note in Appendix C) their method includes additional examples computed from the inner optimization loop. Comparing with other methods that use more inner steps is not a direct comparison. Instead, it would be useful to compare to variants of these methods that, say, compute two adversarial examples in the inner loop and train on the worst one or on both.

**Summary Of The Paper:**

The authors study adversarially robust learning where the goal is to train a model that predicts correctly, even under small, worst-case perturbations of its inputs. The authors focus on “collaborative examples”: inputs close to natural ones that have been perturbed to induce very low loss. These examples are then incorporated into an adversarial training method which the authors find to slightly improve over existing ones. Essentially, the idea is: instead of minimizing the difference in prediction between the original example and its adversarial counterpart (e.g., TRADES), minimize the difference in prediction between any pair of points within the L-p ball around the original example.

**Summary Of The Review:**

The main idea proposed is interesting and does provide non-trivial improvements on a number of datasets. However, the justification and analysis of the method could be significantly improved.

---
Given the additional experimental results provided by the authors, I believe that the analysis is sufficient to support the claims of the paper. I thus increase my score.

---

> ### Author Response · Authors · 2022-11-16
> **Response to Reviewer 84B3**
>
> Thanks for your positive comments. Our response to the comments are provided as follows.
>
> >The existence of such examples is more or less implied by the existence of adversarial examples in the first place (if increasing the loss with small perturbations is easy, one would expect that decreasing the loss would be easy as well).
>
> We agree that the existence of collaborative examples is related to the existence of adversarial examples, which also implies non-flatness of the loss landscape of DNNs. However, our discussions also include: 1) visualization of such examples and show what do they look like for robust and non-robust models, 2) angles between adversarial perturbations and collaborative perturbations for robust and non-robust models, 3) how such examples can be utilized to achieve state-of-the-art adversarial robustness. We believe that these discussions are insightful to the community on a variety of topics. For instance, visualization of collaborative examples and collaborative perturbations demonstrates that mountains with a bluer sky are more likely to be classified into its ground-truth class, by a robust ResNet-18, which may shed new light on model explanation and data augmentation.
>
> > If the goal is to reduce the worst-case error, why does the best-case error have an effect? Is there a fundamental reason or does it happen to help make the loss smoother in practice?
>
> Though not directly imposing restriction on adversarial robustness, bridging the gap between best-case prediction and the benign prediction in fact optimizes the lower bound of the local Lipschitz constant, and the local Lipschitz constant is known to be related to adversarial robustness (the worst-case error) and landscape flatness [1].
>
> By incorporating collaborative examples and adversarial examples jointly into training, ST attempts to obtain loss landscapes that are flat not only at the benign data points, but also at any data points within the $\epsilon$-bounded neighborhood of the benign data points. The lower bound can then become tighter in such a setting. An experiment was performed to somehow compare the flatness of four models: one trained to regularize its best-case error, one trained to regularize its worst case error, one trained to regularize the maximal possible gap (i.e., ST), and one trained as normal. We evaluated the maximal (cross-entropy, CE) loss, the minimal loss, and their gap for all these models. Note that, for inputs that are sufficiently close to each other, the maximal gap between their predictions indicates local flatness of the model landscape, and smaller gap implies flatter loss landscape. Experimental results are provided as below. We see that the regularized models all achieve smaller gaps, while ST achieves the smallest.
>
> |        | Clean CE | Maximal CE | Minimal CE | Gap |
> | ---- | :----:  | :----: | :----:  | :----:  |
> | Normal                           | 0.20 | 24.92 | 0 | 24.92 |
> | Best-case regularized  | 0.54 | 1.69 | 0.23 | 1.46 |
> | Worst-case regularized  | 0.68 | 1.26 | 0.34 | 0.92 |
> | ST                                   | 0.81 | 1.29 | 0.53 | 0.78 |
>
> > It would be useful to compare to variants of these methods that, say, compute two adversarial examples in the inner loop and train on the worst one or on both.
>
> Thanks for the suggestion. We agree that it is a reasonable choice to further compare to previous adversarial training methods in a way that two adversarial examples are computed for each benign sample. We have tried to modify all compared methods, by generating two adversarial examples in the inner loop and regularizing the mean/max of two regularization terms. Nearly all experiments in Section 5 have been re-runned and the results are provided in Appendix C in our paper. Some key conclusions can be summarized as follows: 1) adopting the mean of two regularizations seems slightly more effective than the max, 2) none of the compared methods improve consistently in such an innovative formulation, and TRADES$^\dagger$ performs better than the vanilla AT$^\dagger$ and MART$^\dagger$, 3) combined with AWP or not, in such an innovative formulation involving two examples and regularization from each of them, state-of-the-arts are still inferior to our ST.
>
> &nbsp;
> &nbsp;
> &nbsp;
>
> [1] Hein M, Andriushchenko M. Formal guarantees on the robustness of a classifier against adversarial manipulation. NeurIPS, 2017.

---

> > ### Comment · Reviewer_84B3 · 2022-11-19
> > **Appreciate the additional experiments**
> >
> > I appreciate the authors' effort in providing additional experiments. I actually find it quite interesting that Squeeze Training consistently outperforms other methods, even when these compute two perturbation per inner loop.
> >
> > I do think that the paper would be of interest to the adversarial robustness community, thus I increase my score.

---

> > > ### Author Response · Authors · 2022-11-19
> > > **Thanks for your feedback!**
> > >
> > > Dear Reviewer 84B3,
> > >
> > > We would also like to appreciate the enlightening suggestions. Many thanks!
> > >
> > > Sincerely.

---

### Official Review · Reviewer_Yw6r · 2022-10-29

**Confidence:** 3
**Clarity, Quality, Novelty And Reproducibility:** Pls see the above comments.
**Correctness:** 3
**Technical Novelty And Significance:** 3
**Empirical Novelty And Significance:** 3
**Recommendation:** 6

**Strength And Weaknesses:**

Strength:

1. Using collaborative samples in adversarial training is novel and effective, and it can be used as a plugin for some data augmentation-based adversarial training methods.

2. The proposed ST method is simple and effective, and the theoretical analysis is reasonable and correct.

3. They conduct comprehensive experiments to evaluate the performance of the proposed ST.


Weaknesses:

1. P4, Sec. 3.2, Paragraph 3: Line 1 ‘using the method introduced in Section 3.1. Eq. (5)’ should be Eq. (4).

2. One question is the ratio and sample size of adversarial examples and collaborative examples in ST. Is the total sample size for ST twice the sample size of regular adversarial training or is the sample size the same and adversarial examples and collaborative examples each account for 50%?

3. P8, Sec. 5.1, Table 2: In the comparison between ST and TRADES, the scaling factor values are inconsistent. Can you explain why not adopt the same values of scaling factor, since ST and TRADES are based on similar ideas?

4. The idea of ST in this paper is based on TRADES, and the generation of collaborative examples is based on the reverse operation of PGD. The idea is novel, but the innovation is slightly weak.


**Summary Of The Paper:**

The vulnerability of DNNs to adversarial examples is related to local non-smoothness and the steepness of loss landscapes. To solve the above problem, the main contribution of this paper is that they explore the existence of collaborative examples by simply adapting the PGD method by gradient descent rather than gradient ascent in the original PGD. They propose squeeze training (ST) by considering utilizing collaborative examples and adversarial examples jointly during training to regularize non-smooth regions of the whole loss landscape. Empirically, their method can outperform some SOTA methods under various attacks and datasets.

**Summary Of The Review:**

Pls see the above comments.

---

> ### Author Response · Authors · 2022-11-16
> **Response to Reviewer Yw6r**
>
> Thanks for your positive feedback. Our response to the comments are provided as follows.
>
> >P4, Sec. 3.2, Paragraph 3: Line 1 ‘using the method introduced in Section 3.1. Eq. (5)’ should be Eq. (4).
>
> Thanks for pointing it out. We have revised Section 3.2 accordingly to avoid misunderstanding.
>
> >Question about the sample size and the ratio of adversarial examples and collaborative examples in ST.
>
> For each benign example in the training batch, ST (as introduced in Algorithm 1) generates a pair of two examples for squeezing gaps in its bounded neighborhood. The constraint in Eq. (6) ensures that such a regularization involves one adversarial example and one collaborative example, if possible. While for previous adversarial training methods, only one adversarial example is generated for each benign example. Thus, indeed, the sample complexity of ST is higher. Following the suggestion from Reviewer 84B3, we have tried to innovate previous methods such that two adversarial examples were generated and utilized for each benign example. In such a setting where the sample complexity is all strictly equal, the comparison results still demonstrate remarkable superiority of our ST. All discussions about computational and sample complexity of our method are provided in Appendix C.
>
> >Question about different scale factor values used for TRADES and ST.
>
> TRADES regularizes output discrepancy between benign examples and their neighboring data points, while our ST penalizes the maximum possible output discrepancy between any two data points within the $\epsilon$-bounded neighborhood. With the same choice for the regularization function $\ell_{reg}$, the ST regularization bounds TRADES from above, and it imposes a more significant penalty to landscape flatness with the same scaling factor. Thus, **to ensure that both methods achieve their optimal trade-off** in Figure 6, we adopted larger values for TRADES.
>
> >The idea of ST in this paper is based on TRADES, and the generation of collaborative examples is based on the reverse operation of PGD.
>
> We would like to politely mention that, although the formulation in Eq. (6) is partially inspired by that of TRADES, collaborative examples may in fact be incorporated into other adversarial training formulations to improve their performance in a similar spirit. Moreover, in order to encourage the pair of examples involving one adversarial example and one collaborative example, a chained equality constraint should be introduced and the optimization process has been innovated accordingly, making the generation of collaborative examples and adversarial examples different from the conventional PGD. See Algorithm 1 and discussions in Section 4.1 for details.

---

### Public Comment · ~Wenhai_Wan1 · 2023-06-28
**What's the meaning of 'Original Prob' and 'Max Prob' in Fig3?**

as the title described.

---

### Decision · Program_Chairs · 2023-01-20

**Decision:**

Accept: poster

**Justification For Why Not Higher Score:**

The motivation of the method e.g. regarding improving local Lipschitz constants could be improved. While the method seems to lead to consistent improvements, the gains in performance are small.

**Justification For Why Not Lower Score:**

The experiments look solid (strong attacks like AutoAttack are used for evaluation), the improvements are consistent over three datasets. The method is simple and efficient and seems even outperform existing methods when one considers the doubled computational effort.

**Metareview: Summary, Strengths And Weaknesses:**

The paper suggest that during adversarial training one should not penalize just the "worst case adversarial example" (lowest confidence in correct class) but the difference of worst and best case (highest confidence in correct class in the ball around the target example). While this doubles the computational effort during training (two attacks have to be done simultaneously), the authors show that this consistently improves over other adversarial training approaches (in particular TRADES which is closest to the suggested approach).

Strength:
- interesting, novel and simple modification of TRADES which is effective
- experimental gains are small but seem to be consistent. The new results which take into account the extra work of the suggested method still show an advantage (Table 6)

Weakness:
- there is little motivation for the method, the theoretical connection to the local Lipschitz constant could be improved (in one answer in the rebuttal it looks like that the local Lipschitz constant gets better but it is not directly measured)
- the experimental gains are small and for some runs (Table 3/4) no standard deviation is reported

The authors could clarify all doubts and questions of the reviewers, in particular regarding a fair comparison to existing approaches due to their two times increased computational effort. All reviewers recommend acceptance of the paper.

For the final version I suggest to:
- include the experiments with fair comparison regarding computational cost in the main paper
- report local Lipschitz constants for different AT alternatives
- check effect on other threat models like $l_2$

**Note From Pc:**

if the above contains the word "oral" or "spotlight" please see: "oral" presentation means -> notable-top-5% and "spotlight" means -> notable-top-25%. As stated in our emails, we are disassociating presentation type from AC recommendations